# Eye Tracking Post Processing to Detect Visual Artifacts and Quantify Visual Attention under Cognitive Task Activity during fMRI

**DOI:** 10.3390/s24154916

**Published:** 2024-07-29

**Authors:** Maxime Leharanger, Pan Liu, Luc Vandromme, Olivier Balédent

**Affiliations:** 1CHIMERE UR 7516, Jules Verne University of Picardy, 80000 Amiens, Franceluc.vandromme@u-picardie.fr (L.V.); 2Medical Image Processing Department, Amiens Picardy University Medical Center, 80000 Amiens, France

**Keywords:** eye-tracking, cognitive tasks, visual stimuli, behavioral feedback, signal processing

## Abstract

Determining visual attention during cognitive tasks using activation MRI remains challenging. This study aimed to develop a new eye-tracking (ET) post-processing platform to enhance data accuracy, validate the feasibility of subsequent ET-fMRI applications, and provide tool support. Sixteen volunteers aged 18 to 20 were exposed to a visual temporal paradigm with changing images of objects and faces in various locations while their eye movements were recorded using an MRI-compatible ET system. The results indicate that the accuracy of the data significantly improved after post-processing. Participants generally maintained their visual attention on the screen, with mean gaze positions ranging from 89.1% to 99.9%. In cognitive tasks, the gaze positions showed adherence to instructions, with means ranging from 46.2% to 50%. Temporal consistency assessments indicated prolonged visual tasks can lead to decreased attention during certain tasks. The proposed methodology effectively identified and quantified visual artifacts and losses, providing a precise measure of visual attention. This study offers a robust framework for future work integrating filtered eye-tracking data with fMRI analyses, supporting cognitive neuroscience research.

## 1. Introduction

The use of functional MRI (fMRI) allows for an in-depth study of the correlations between eye movements and brain activations, especially in psychological domains such as joint attention, which is crucial for cognitive and social development [1,2]. Joint attention refers to the ability of an individual to share an event with another person by directing their attention toward a person or an object, thus forming a triadic environment [3,4]. This process, which develops early in childhood, is essential for social cognition and involves sequences of mutual and directed gazes between two people and a target object. Studies on joint attention have shown activations in various brain regions, including the dorsomedial prefrontal cortex (dmPFC) for social partner perception, the superior temporal sulcus (STS) for detecting attention shifts, the frontal eye fields (FEF) and temporoparietal junction (TPJ) for eye movement control, and the intraparietal sulcus (IPS) for spatial attention [5,6].

Functional MRI provides excellent spatial resolution, allowing for detailed mapping of brain areas involved in various tasks. This technique is based on the Blood Oxygenation Level Dependent (BOLD) signal, which reflects changes in blood flow to different brain regions, thus highlighting areas of activation [7]. However, fMRI has a relatively poor temporal resolution, with acquisition times typically on the order of seconds, necessitating the repetition of stimuli and prolonged scanning sessions to capture dynamic processes [8]. Additionally, the BOLD signal has an inherent temporal delay of several seconds, complicating the precise correlation between neural activity and observed behavior [9]. The BOLD signal is also relatively weak, requiring multiple repetitions of stimuli and block paradigms to enhance statistical power [10]. Block paradigms, where periods of task performance alternate with rest periods, help increase the signal-to-noise ratio, but can limit the study of more complex, dynamic cognitive processes [11].

These limitations highlight the need for complementary techniques like eye tracking, which offers high temporal resolution, capturing eye movements on the order of milliseconds [12]. Combining eye tracking with fMRI allows researchers to align high-temporal-resolution behavioral data with high-spatial-resolution neural data, providing a more comprehensive understanding of cognitive processes [13,14]. Eye trackers measure where and how long a person looks at various points in their visual field, capturing both fixation (when the gaze is held steadily on a single location) and saccades (quick eye movements between fixations) [15]. The hardware component of eye tracking systems typically involves infrared light sources and cameras. Infrared light is directed toward the eyes, and the cameras capture reflections from the cornea and the retina. These reflections are used to calculate the point of gaze by triangulating the positions of the reflections and the center of the pupil [16]. The accuracy and precision of these measurements depend on the quality of the hardware, such as the resolution of the cameras and the processing power of the system. Modern eye trackers can achieve a high degree of accuracy, often less than 0.5 degrees of visual angle, and can sample gaze positions at rates exceeding 1000 Hz, allowing for detailed tracking of rapid eye movements [12].

Software algorithms play a crucial role in processing raw eye-tracking data to identify fixations and saccades. Several algorithms have been developed, each with its strengths and limitations. Common methods include the Velocity-Threshold Identification (I-VT), which classifies eye movements based on their speed; the Dispersion-Threshold Identification (I-DT), which uses the spatial dispersion of gaze points, and more sophisticated models like the Hidden Markov Model Identification (I-HMM) and the Kalman Filter Identification (I-KF), which incorporate probabilistic models of eye movement behavior [17,18]. The choice of algorithm can significantly affect the interpretation of eye-tracking data, as different methods may yield varying results for the same raw data. For instance, I-VT is known for its simplicity and real-time processing capabilities, but it may overestimate the number of saccades in noisy datasets. The I-DT algorithm addresses some of these limitations by incorporating spatial dispersion criteria, which can better distinguish between fixations and saccades in noisier datasets. However, it may miss shorter fixations due to its reliance on spatial thresholds [18]. More advanced algorithms like I-HMM and I-KF provide sophisticated modeling of eye movement dynamics, offering greater accuracy in fixation and saccade classification, but their computational demands can limit their real-time applicability.

One critical limitation of these algorithms is that the parameter settings for these classifiers are often based on data recorded from individuals without any psychological or neurological disorders. This can be problematic, as certain conditions can affect eye movement patterns. For example, individuals with autism spectrum disorder (ASD) are known to exhibit more peripheral vision and less central gaze fixation, which can lead to inaccuracies in fixation classification using standard algorithms [3,19,20]. Similarly, increased blink rates are observed in conditions like ADHD, which can interfere with the accurate classification of saccades and fixations [21]. These examples illustrate how the use of standard parameter settings may not be suitable for all populations, leading to potential misinterpretations in studies involving individuals with unique visual attention profiles. These variations in algorithm performance make it challenging to compare results across different studies, as the same raw data might yield different outcomes, depending on the chosen method. This inconsistency is particularly problematic when studying populations with unique visual attention profiles, such as individuals with visual attention disorders.

Moreover, integrating eye tracking with fMRI presents additional challenges, such as performing drift checks before each trial to minimize motion artifacts and adjusting calibration methods to compensate for drifts caused by eye dryness or participant movement [22]. Furthermore, to increase statistical power and compensate for the low signal-to-noise ratio of fMRI, repeated block paradigms are essential [7,8,10]. However, these repeated tasks can lead to participant fatigue, potentially reducing engagement and vigilance over prolonged sessions. Currently, there is a lack of research on the relationship between participant engagement and session duration [11]. Furthermore, drift checks, often imposed before each trial to enhance eye-tracking accuracy, introduce additional tasks that are not directly related to the experimental paradigm, but are necessary to maintain calibration. This can influence the interpretation of results by adding a layer of visual stimuli unrelated to the primary task [22].

In fMRI data processing, tools like SPM12 utilize the General Linear Model (GLM) to determine the timing of stimulus presentations during scanning sessions, allowing researchers to specify onset times and conditions of interest [23]. However, selecting volumes based on pre-defined paradigms can introduce biases in the interpretation of results. For instance, in studies of joint attention, researchers typically compare activation phases (joint attention) against baseline phases (no joint attention) and predefined regions of interest, which may not fully account for the participants’ actual behavior during the scan [24]. This is especially critical when studying populations with neurodevelopmental disorders, where task performance may deviate significantly from the expected patterns. To create a more naturalistic setting, researchers can minimize explicit instructions, thus allowing for spontaneous interactions that are more reflective of real-world scenarios [25]. For example, in the study by Oberwelland et al. [24], volumes were selected based on participants’ actual gaze behavior, enhancing the ecological validity of the findings. This methodology underscores the importance of using raw eye-tracking data to avoid biases from predefined classification algorithms, and allows for more direct correlations with fMRI data, ultimately providing a richer understanding of the neural mechanisms underlying visual attention and social interaction [2,23].

The objectives of this study are outlined as follows:

**Develop adjustable filtering algorithms**: The first objective is to create advanced algorithms capable of removing sensor drift noise and blink artifacts from raw eye-tracking data. These enhancements aim to improve the accuracy of engagement quantification, which is essential for reliable eye-tracking analyses.

**Explore changes in participant engagement over time**: The second objective is to investigate the variations in participant engagement over the course of the experiment. Understanding these changes will help determine the optimal duration for data collection, ensuring that high levels of participant attention are maintained. This is particularly crucial for future applications of eye-tracking in fMRI studies, where maintaining consistent engagement is key to obtaining valid and reliable results.

By addressing these objectives, this study seeks to enhance the methodologies used in eye-tracking research, thereby providing a robust framework for future studies that integrate eye-tracking and fMRI techniques.

## 2. Materials and Methods

### 2.1. Participants

A sample of 16 neurotypical participants aged 18 to 20 years was recruited for this study. Written informed consent was obtained from all participants. Inclusion criteria also included the absence of contraindications for MRI. Ethical approval was provided by the appropriate authority, and the corresponding ethical approval code is [PI2021_843_0194].

Participation in this study does not entail any particular risk. MRI examination presents no risk to the subject once contraindications are ruled out: it is non-invasive, non-ionizing, and non-irradiating. Similarly, eye tracking (ET) is non-invasive and emits infrared light that does not affect human retinal cells.

The exclusion criteria were:Age under 18 years.Presence of MRI contraindications, including:
-Claustrophobia;-Pacemaker, neurosurgical clips, vascular clips, cardiac valves, ventriculoperitoneal shunts, cochlear implants, neurostimulators, intraocular metallic fragments, joint prostheses.Severe obesity (>140 kg) preventing entry into the MRI tunnel (diameter < 70 cm).Pregnant or breastfeeding women.Individuals under guardianship or curatorship, under judicial protection, or deprived of liberty by judicial or administrative decision.Individuals with psychiatric disorders such as attention deficit disorder with or without hyperactivity, depression, bipolar disorder, and schizophrenia.Individuals with neurological history such as epilepsy and/or cerebrovascular accident.Metal workers or individuals with intraocular foreign bodies.

### 2.2. Experimental Task

Participants performed a joint attention (AJ) task during the fMRI acquisition. The task involved viewing videos where an interlocutor (an actress’ face) made eye movements towards objects. The task had two conditions (Figure 1):**Joint Attention (AJ)**: The interlocutor initially gazed straight ahead, then oriented their gaze towards one of two surrounding objects (cheese slices) where a mouse appeared. Each video sequence was repeated for a total of 25 s per block. In each video, the interlocutor (an actress) initially fixed their gaze straight ahead, then performed an eye movement towards one of the two surrounding objects (eye movement starting at 1 s), randomly (equal number of trials to the right and left throughout the experiment). A mouse appeared at the cheese slice the interlocutor fixated on. At the end of the video, the interlocutor redirected their gaze straight ahead (returning their gaze at 4 s of the video).**Without Joint Attention (NOAJ)**: Similar to the AJ block, but without the interlocutor’s eye movement.

These blocks were repeated six times, with periods of fixation on a cross at the center of the screen for 20 s between each block, totaling 280 s per session. In total, there were 12 blocks (4 rest, 4 activation, and 4 with a cross) and 44 trials (20 rest, 20 activation, and 4 with the cross to verify drift). The sequence of these blocks was as follows: 25 s AJ (5 videos), 25 s NOAJ (5 videos), and 20 s cross (image of a cross).

### 2.3. Eye Tracking Data Acquisition

The development of the “Human Interactions and Eye-Tracking” technical platform at UPJV’s Digital Humanity platform recently enabled the acquisition of a high-performance MRI-compatible eye tracker EyeLink 1000 Plus (SR Research, Mississauga, ON, Canada), allowing simultaneous recording of gaze and brain activation mapping by fMRI (Siemens, Munich, Germany). The eye tracker was installed at the back of the MRI tunnel, outside the 5 Gauss line. An experimental protocol to test visual activity during the exploration of social percepts, considering the constraints of fMRI, which requires long acquisition sequences and precise MRI-ET synchronization (every 2500 ms), was developed. The Experiment Builder software (SR Research) was set to 1000 Hz on one eye, obtaining 1000 eye position measurements per second. The ET algorithm categorizes anything not identified as a saccade or blink as a fixation. The screen sampling rate was 60 Hz with a resolution of 1680 × 1050.

### 2.4. Participant Preparation and Instructions

Before entering the MRI machine, the investigator explained to the participants that they would be performing a social interaction task live on a screen (displayed on the MRI monitor, SensaVue Patient Display, Invivo, Gainesville, FL, USA). First, a calibration phase was conducted to set the gaze position (monocular) relative to the screen. Participants were instructed to precisely fixate on circles appearing and disappearing randomly at 9 positions on the screen (grid pattern). A precision of less than 0.5°and an accuracy of less than 1° were considered acceptable for calibration. If calibration failed, it was repeated. A vision test was also offered to verify the participants’ vision at a distance of 1 m. If the vision test failed, plastic glasses were provided.

Participants lying in the MRI machine viewed the screen via a tilted mirror above their heads. Participants viewed the screen through a mirror within the head coil, with a path length of 1 m from the screen to their eyes. The instruction was to fixate on the cross as accurately as possible during the cross-fixation block and to freely view the screen for the rest of the task (Figure 2a,b).

### 2.5. Signal Post-Processing

We conducted preliminary tests to evaluate the potential effects of eye-tracking equipment on fMRI image quality using a phantom study. These findings confirm that the equipment did not influence the quality of the fMRI images (Figure A1 and Figure A2).

Then, this signal processing procedure is completed using an Excel spreadsheet based on macros (Microsoft, Redmond, WA, USA).The eye movement sensor data primarily contains three types of errors (Figure 2c). Two are caused by the sensor, and one is caused by the subject’s blinking:**Transient spike**: a brief shift in eye movement data that causes the gaze point to temporarily deviate from the correct position and then quickly return to the correct trajectory, usually lasting a few milliseconds.**Spatial displacement**: A short-term displacement of the recorded eye movement data to an incorrect position with the sensor continuing to record the incorrect position for a period. However, the relative displacement is accurate before returning to the correct position, generally lasting from several tens to a few hundred milliseconds.**Blink artifact**: the loss of eye movement data during a subject’s blink causing the fixation point to move sharply up or down for about 0.1 s before and after the blink, which affects the final data.

First, for the two types of errors caused by the sensor, we need to define an eye movement threshold. This threshold can be determined by converting the sensor’s values into units of eye movement speed, i.e., degrees per second (°/ms) based on the distance between the eyes and the screen, as well as the screen size (Figure 2a,b). This threshold is set according to the upper limit of saccade speed (0.9°/ms, i.e., 40 unit/ms) [26,27], and can be manually adjusted. Additionally, since the direction of fixation point movement is opposite at the start and end of these errors, this characteristic can help define the intervals of erroneous data. By setting the eye movement speed threshold and utilizing the characteristics of movement direction, we can accurately identify and filter these erroneous data.

The third type of error is characterized by the representation of position data in the time series by null values. This allows for the identification of the moments and durations of blinks. The following correction schemes are applicable to the three types of errors (Figure 2d):**Transient spike**: since it usually lasts for a short duration, directly replace the erroneous data with the correct data from the final moment.**Spatial displacement**: due to its longer duration and the ability to record the correct trajectory (despite the positional error), use the first derivative of the position (displacement/ms) to calculate the correct position at each moment until the gaze point returns to the correct position.**Blink artifact**: Delete the entire interval of approximately 0.2 s before and after the blink (during which the gaze point moves sharply up or down). Use linear interpolation to fill this blank area.

Finally, a specialized protocol is employed to verify the efficacy of this post-processing program. The validation protocol comprises two distinct components:Subjects are instructed to fixate on the center of the screen while continuously blinking (Figure 3).Subjects are required to scan a spiral line on a background board from the inside out (Figure 3).

As illustrated in Figure 3, the post-processed data exhibits a clear improvement in the aforementioned three types of error.

### 2.6. Group Analysis

In this study, we analyzed gaze positions for each condition (“Center”, “AJ”, “NOAJ”) to evaluate the effectiveness of filtering on eye-tracking data. For each condition, gaze positions were categorized as either “original” (ori) or “filtered” (filted) based on the processing described in Section 2.5. The analysis focused on three main aspects: overall screen positions, task-specific gaze positions, and the temporal effect on task performance.
**Calculation of individual means for screen positions**: For each participant, we calculated the mean number of gaze positions that fell within the screen boundaries (0 to 1680 in X coordinates and 0 to 1050 in Y coordinates) for each condition (“Center ori”, “AJ ori”, “NOAJ ori”) and their corresponding filtered data (“Center filted”, “AJ filted”, “NOAJ filted”). These individual means were then averaged across all participants to obtain the overall means for each condition. These group-level means were used to perform statistical comparisons between the original and filtered data sets. All calculations of means were performed using Excel.**Calculation of task-specific gaze positions**: For each condition, we also calculated the number of gaze positions, indicating task adherence. This involved counting the number of gaze positions on the fixation cross during the “Center” condition and the number of gaze positions on target objects during the “AJ” and “NOAJ” conditions. These calculations were based on predefined areas of interest (AOIs) around the cross and target objects. For each participant, we computed the mean number of gaze positions within these AOIs for both the original and filtered data of each condition.**Temporal analysis of task performance**: To assess the effect of time on task performance, we calculated the mean number of task-specific gaze positions for each trial of each condition for every participant. These trial-wise means were then used to perform a non-parametric Friedman test on the filtered data to determine if there were significant temporal effects on task performance. This step allowed us to evaluate whether participants’ adherence to task instructions remained consistent over time.

The non-parametric Wilcoxon signed-rank test was used to assess the differences between the original and filtered data for each condition. All tests were two-tailed, with significance set at *p* < 0.05. Using the non-parametric Friedman test combined with post hoc Wilcoxon signed-rank tests, we assessed the impact of time on participant attention. Bonferroni correction was applied to control for the errors arising from multiple comparisons. All statistical comparisons and figures were performed using Python via the Spyder IDE (Anaconda, Inc., Austin, TX, USA) and R with Rstudio IDE (RStudio, PBC, Boston, MA, USA).

## 3. Results

### 3.1. Comparison of Gaze Positions on the Screen before and after Filtering

The means and standard deviations of the gaze positions of the participants on the screen for each condition, before and after filtering, are summarized below. For the “Center ori” condition, the mean gaze position was 0.975 with a standard deviation of 0.049, whereas after filtering (“Center filted”), the mean increased to 0.999 with a reduced standard deviation of 0.003. In the “AJ ori” condition, the mean was 0.900 with a standard deviation of 0.074, and it improved to a mean of 0.943 with a standard deviation of 0.078 after filtering (“AJ filted”). Similarly, for the “NOAJ ori” condition, the mean gaze position was 0.891 with a standard deviation of 0.087, and after filtering (“NOAJ filted”), the mean increased to 0.947 with a standard deviation of 0.088.

As shown in Table 1, compared to the raw data, the proportion of time spent fixating on the screen increased in the filtered data across all three conditions (*p* < 0.001). The mean gaze positions ranged from 89.1% to 99.9%.

### 3.2. Comparison of Gaze Positions on the Task before and after Filtering

The gaze positions of the participants on the task (i.e., during the cross condition and during the AJ and NOAJ conditions) were also analyzed. For the “Center ori” condition, the mean was 0.704 with a standard deviation of 0.268, and it slightly improved to a mean of 0.723 with a standard deviation of 0.266 after filtering (“Center filted”). In the “AJ ori” condition, the mean was 0.462 with a standard deviation of 0.185, increasing to a mean of 0.477 with a standard deviation of 0.196 after filtering (“AJ filted”). For the “NOAJ ori” condition, the mean gaze position was 0.473 with a standard deviation of 0.208, and it improved to a mean of 0.500 with a standard deviation of 0.220 after filtering (“NOAJ filted”).

These results show that participants generally performed the task as instructed, with gaze positions ranging from 46.2% to 50% for each condition. The consistency in data dispersion is indicated by the close standard deviations before and after filtering.

Table 2 presents the mean rates of gaze positions on the task for each condition before and after filtering, as well as the corresponding standard deviations. These results indicate significant improvements in gaze positions on the task after the filtering process.

### 3.3. Effect of Time on Gaze Positions Depending on the Task and Conditions

The analysis revealed statistically significant differences in the Eye Cross between the first and fifth trials (*p*-value = 0.0054), but not in the AJ (*p*-value = 0.326) and NOAJ (*p*-value = 0.536) trials.

Figure 4 shows the effect of time on the position of the gaze during the cross condition, along with the corresponding standard deviations. Error bars and significance symbols are also included to illustrate the statistically significant differences between the pairs of trials.

## 4. Discussion

By initially working with a homogeneous group of young, healthy adults, we ensured that our methodology is robust and reliable before applying it to more complex and variable populations. Future research will aim to include a more diverse demographic to provide comprehensive insights and validate our findings across different populations, including individuals with Alzheimer’s disease, attention deficits, and neurodevelopmental disorders. These populations present unique challenges in interpreting brain activation maps due to potential variations in patient participation and engagement with the stimuli.

### 4.1. Eye-Tracking Accuracy and Task Adherence

The significant differences observed between the original and filtered gaze position data across all conditions underscore the effectiveness of the filtering process. The percentages indicated reflect the proportion of gaze directed at the screen. For instance, the mean gaze position accuracy improved from 0.975 to 0.999 in the “Center” condition and from 0.900 to 0.943 in the “AJ” condition post-filtering. This enhancement in data quality is consistent with findings from prior research, demonstrating that advanced signal processing techniques can significantly reduce noise and improve the reliability of eye-tracking data [12,28]. These data show that the participants were not looking off-screen; rather, the increase in fixation on the central cross, discussed in later sections, indicates a resolution of sensor-related issues, aligning with participant instructions. This demonstrates that the observed improvements in gaze accuracy are attributable to the filtering process, which effectively addressed sensor errors, thus reflecting true participant behavior [15,18,29].

The significant improvements in gaze position post-filtering, from a mean of 0.462 to 0.477 in the AJ condition and from 0.473 to 0.500 in the NOAJ condition, support the hypothesis that eye-tracking can provide valuable real-time assessments of participant engagement. The percentages in the AJ and NOAJ conditions are lower than those required by the task of fixating on the cross, as the mouse appears in one of the surrounding objects for 3 out of the 5 s of each video (3/5 of the time). Moreover, the adherence to task-specific gaze positions improved post-filtering, indicating that participants generally performed the tasks as instructed. This observation supports the hypothesis that eye-tracking can effectively monitor and ensure participant compliance in cognitive tasks, which is crucial for the validity of fMRI studies. The improvement in mean gaze positions on specific tasks, such as the cross-fixation and AJ conditions, further validates the use of eye-tracking as a complementary tool to fMRI [17,30,31].

Similarly, Siegmund et al. [32] explored the feasibility of simultaneously measuring program comprehension with fMRI and eye tracking. They faced challenges with eye-tracking data quality due to the fMRI environment, where the high blink rate and calibration difficulties led to a high failure rate in capturing eye movements consistently. In contrast, our study successfully mitigates these issues by implementing advanced post-processing techniques, resulting in higher data accuracy. For instance, Siegmund et al. reported that their eye tracker captured less than 30% of the experiment time for some participants. Moreover, the spatial error in Siegmund et al.’s study impacted the detection of fixations on individual identifiers, while our approach effectively identified and quantified visual artifacts, ensuring precise measures of visual attention.

In summary, the comparison underscores the importance of robust post-processing algorithms in enhancing eye-tracking data quality, especially in complex fMRI environments. Our findings provide a more reliable framework for integrating eye-tracking with fMRI, supporting cognitive neuroscience research and offering valuable tools for future studies.

### 4.2. Temporal Consistency and Cognitive Engagement

The analysis of temporal effects on task performance revealed significant differences in the Eye Cross trials but not in the AJ and NOAJ trials.

We acknowledge the concern regarding participant fatigue during prolonged scanning sessions and repeated tasks. Our study aimed to assess whether participants adhered to instructions over time, such as maintaining gaze on a central cross. We observed a temporal effect on this condition, as shown in Figure 4. However, no significant temporal effects were found in other conditions, likely due to the cognitive engagement required for tasks involving joint and non-joint attention. Our study results are consistent with previous findings [33,34,35], showing that involuntary eye movements persist even when focusing on a single object. This can lead to fatigue during prolonged fixation tasks, causing cognitive engagement and attention to decline over time, especially in tasks requiring sustained attention. Despite the instruction to freely view the screen during these conditions, participants still performed the joint attention task, highlighting the natural inclination for joint attention in neurotypical individuals. It is important to maintain gaze percentage as a measure of task performance, as this indicator can reveal the difficulties that individuals with neurodevelopmental disorders may encounter [12,15].

Marsman et al. [36] use of fixation-based event-related fMRI analysis to reveal cortical processing during free viewing of visual images. They highlighted the potential of using fixations as events in fMRI to study cortical processing. Our approach, focusing on high-quality raw data, aligns with their findings by ensuring a high temporal resolution and avoiding the biases that can arise from different eye-tracking classification methods. In line with this, Murphy and Garavan [37] highlighted that brain activation maps can differ significantly based on individual performance, which is crucial for interpreting activation maps. The simultaneous use of eye-tracking with fMRI allows for performance-based adjustments in activation, particularly in cognitive tasks where gaze and visualization are heavily utilized. This method ensures that brain activation patterns reflect actual behavioral performance, enhancing the reliability of fMRI results. Additionally, Rusch, in 2021 [38], emphasized the integration of fMRI and eye-tracking for studying social cognition, noting that fixation and saccade data are often used as explanatory variables in fMRI analyses, usually as parametric modulators. However, in our study, we focused on high-quality raw data due to its high temporal resolution, afforded by a sampling rate of 1000 Hz. This approach avoids the variability introduced by different classification algorithms for fixations and saccades used by various eye-tracking manufacturers and experimental setups. Furthermore, Oberwelland et al. [39] used an interactive eye-tracking paradigm combined with fMRI to investigate the developmental trajectories of joint attention (JA). They found that JA activates similar networks of “social brain” areas in children and adolescents as in adults, highlighting the importance of the temporoparietal junction and precuneus in these processes. Our study demonstrated the feasibility and accuracy of eye-tracking in conjunction with fMRI, reinforcing the value of robust post-processing techniques.

### 4.3. Limitations and Perspectives

One limitation of this study is the need for manual parameter adjustment in the filtering algorithm, which requires a certain level of prior knowledge. The robustness and reliability of eye-tracking data post-processing could be significantly enhanced by incorporating more parameters or utilizing artificial intelligence for automatic parameter tuning [40,41].

Linear interpolation, used to fill in gaps caused by blinks, can lead to overfitting and potentially introduce artifacts into the data. Previous research has highlighted the limitations of linear interpolation, noting that it can oversimplify the natural variability in eye movement behavior, leading to less accurate representations of gaze dynamics during rapid movements or blinks [17,42,43]. Furthermore, regression techniques to compensate for missing data can sometimes result in values that do not accurately reflect true gaze behavior, especially during high-frequency gaze shifts. While the filtering process enhances data quality, it may introduce biases. Future improvements could involve more sophisticated interpolation methods, such as those based on machine learning algorithms, which can better model the nonlinear nature of eye movements [44,45,46]. Additionally, studies have shown that the choice of eye-tracking hardware and software can significantly impact data quality [18,28,47].

In addition to these considerations, it is important to compare our filtering technique with other common methods used by eye-tracking system manufacturers. These manufacturers often use gap fill-in interpolation, noise reduction techniques like moving average and median filters, and drift correction to enhance the quality of eye-tracking data. While these methods help maintain data continuity and reduce noise, they may introduce artifacts and overfit the data. For example, gap fill-in interpolation fills in missing data caused by blinks or tracking problems, creating a smooth and continuous dataset. Noise reduction techniques, such as moving average and median filters, smooth out noise while preserving essential features needed for fixation classification. Drift correction handles small drifts in gaze position that can build up over time, ensuring the gaze data remains accurate throughout the experiment. However, these techniques may not fully account for the complex, non-linear nature of eye movements, and can introduce biases based on parameter settings. In contrast, our approach focuses on processing raw eye-tracking data without pre-categorizing it into fixations or saccades. We do not exclude or hide data points but recalibrate gaze positions based on the expected positions during blinks, maintaining a high sampling rate and data integrity. This method avoids the pitfalls of introducing saccadic artifacts due to blinks and ensures the data remains as close to its original form as possible. Furthermore, our method includes adaptive gaze correction based on visual angle offset, providing better adjustment and accuracy of corrected gaze data. Unlike other methods that might exclude certain data points, our approach retains high-frequency data and recalibrates gaze positions based on assumed positions during blinks, ensuring the sampling rate remains high and data integrity is maintained. Manufacturers typically use filters without allowing researchers full control over their correction methods, often basing their methodologies on ocular physiology for blink detection. These parameters are neither available nor modifiable, and no gold standard is defined for these detections and corrections. This is why we conducted pre-tests and controlled acquisitions of artifacts to validate our method [22,31,44,45].

## 5. Conclusions

In conclusion, the filtering algorithm developed in this study successfully improves the accuracy of eye-tracking data to a certain extent, and validates that participants’ attention (AJ and NOAJ) does not significantly decline over a 4.6 min collection period. These findings provide valuable tools and data references for the feasibility of integrating eye-tracking technology with fMRI applications.

## Figures and Tables

**Figure 1 sensors-24-04916-f001:**
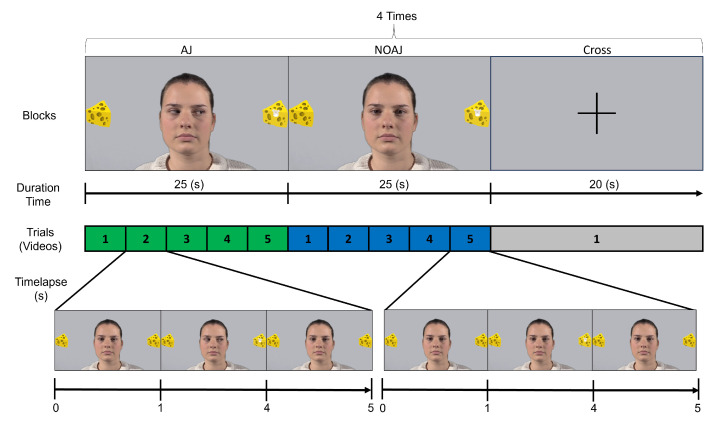
Sequence of the blocks during the experimental task.

**Figure 2 sensors-24-04916-f002:**
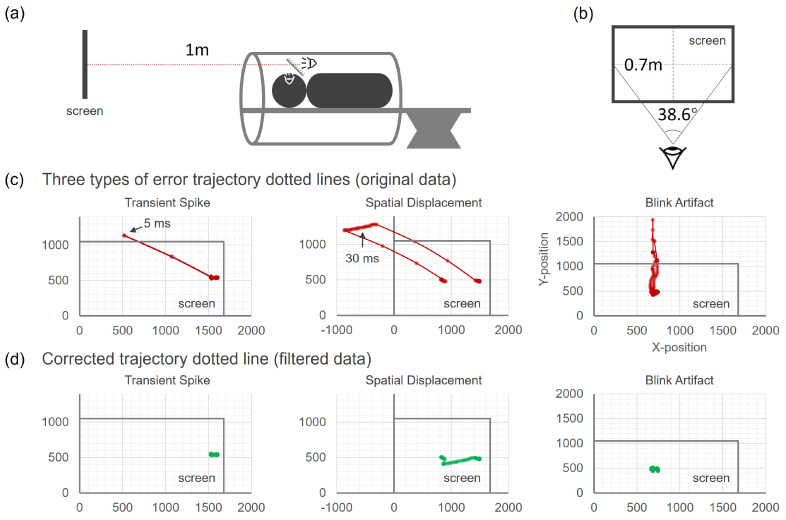
Eye movement data collection and errors: (**a**) Side view of eye movement data collection from the subject. (**b**) The angular range of eye gaze on the screen. (**c**) Dotted-line trajectory under three types of errors: Transient Spike, Spatial Displacement, and Blink. (**d**) Corrected dotted-line trajectory.

**Figure 3 sensors-24-04916-f003:**
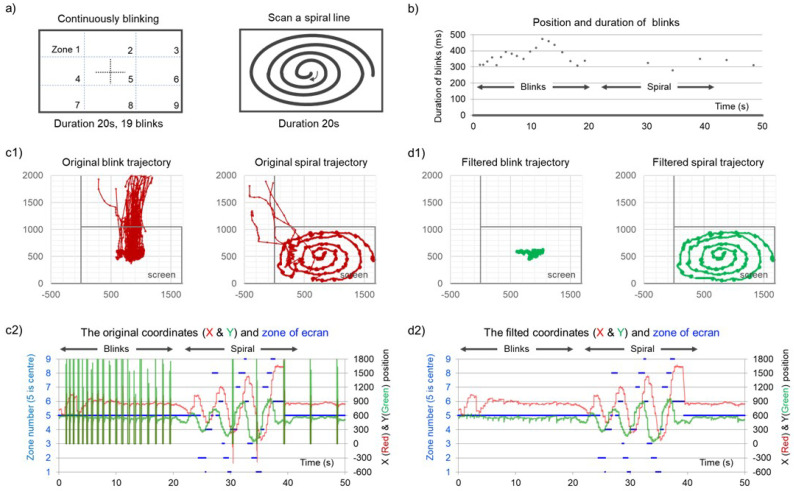
Two 20 s test protocols. (**a**) The subject fixates on the center of the screen and blinks continuously, approximately once per second, then the subject scans a spiral trajectory from the inside out. (**b**) Shows the moments and durations of blinks during the data collection process. (**c1**) Displays the raw path trajectories under the two protocols, while (**d1**) shows the filtered trajectories. (**c2**) and (**d2**) present time-position plots of the raw and filtered data, respectively, where blue indicates the current gaze point position on the screen, which is divided into nine sections as shown in (**a**), and red and green represent the coordinates of the gaze point.

**Figure 4 sensors-24-04916-f004:**
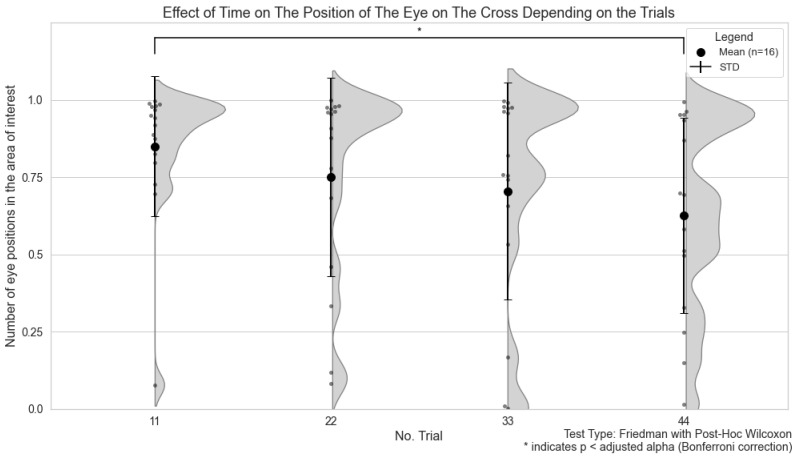
Effect of time on the gaze position during the Central Fixation (Center) condition, where participants fixated on a cross. The figure shows the comparison of gaze positions across trials, highlighting any temporal effects on gaze stability.

**Table 1 sensors-24-04916-t001:** Comparison of mean rates of gaze positions on the screen before (“ori”) and after filtering (“filted”) for each condition: Joint Attention (AJ), Without Joint Attention (NOAJ), and Central Fixation (Center). Values are rounded to the nearest thousandth.

Condition	Original Mean (SD)	Filtered Mean (SD)	*p*-Value
Center	0.975 (0.049)	0.999 (0.003)	<0.0001
AJ	0.900 (0.074)	0.943 (0.078)	<0.0001
NOAJ	0.891 (0.087)	0.947 (0.088)	<0.0001

**Table 2 sensors-24-04916-t002:** Comparison of gaze positions on the task before (“ori”) and after filtering (“filted”) for each condition: Joint Attention (AJ), Without Joint Attention (NOAJ), and Central Fixation (Center). In the AJ condition, participants followed the gaze of an interlocutor towards objects. In the NOAJ condition, the interlocutor did not move their gaze. The Center condition involved fixation on a central cross. Values are rounded to the nearest thousandth.

Condition	Original Mean (SD)	Filtered Mean (SD)	*p*-Value
Center	0.704 (0.268)	0.723 (0.266)	0.001
AJ	0.462 (0.185)	0.477 (0.196)	0.023
NOAJ	0.473 (0.208)	0.500 (0.220)	0.002

## Data Availability

The data presented in this study are available on request from the corresponding author. The data are not publicly available due to data privacy.

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
