# Peer review of "Eye Tracking Post Processing to Detect Visual Artifacts and Quantify Visual Attention under Cognitive Task Activity during fMRI"

_sensors, 2024, doi:10.3390/s24154916_

Round 1

Reviewer 1 Report

Comments and Suggestions for Authors

The submission “Eye-tracking Methodology to Evaluate Subject Active Participation during Activation MRI Cognitive Tasks” proposes a scheme to develop and validate an eye-tracking (ET) methodology integrated with activation MRI to quantify subject engagement during cognitive tasks. Various limitations that could impact the outcomes have been addressed:

·         The problem statement is blurred, and not clear.

·         The achieved objectives do not reflect a real contribution, they are just a direct application.

·         The study is limited to very tied category regarding age (18 to 20 years). This limits the generalizability of the findings to broader populations, including those with psychological or neurological disorders, who may exhibit different eye movement patterns.

·         The study did not consider the potential effects of eye-tracking equipment on fMRI image quality or the impact of fMRI noise on eye-tracking data stability and participant attention.

·         Prolonged scanning sessions and repeated tasks can lead to participant fatigue, which may reduce engagement and vigilance over time.

·         The linear interpolation method used to fill in gaps caused by blinks can lead to overfitting and the introduction of artifacts, potentially misrepresenting true gaze behavior.

Unfortunately, my recommendation is not to accept the paper.

Reviewer 2 Report

Comments and Suggestions for Authors

The main question addressed by this research is whether an eye-tracking methodology, combined with filtering algorithms, can improve the accuracy of data on subject engagement during activation MRI cognitive tasks. The originality of this research lies in its integration of eye-tracking technology with MRI to assess participant engagement during cognitive tasks. This approach addresses the gap in accurately quantifying subject engagement and reducing noise in activation MRI data, which is crucial for the reliability of cognitive task assessments. Including a more diverse demographic, in terms of age and other factors, could also provide more comprehensive insights. Additionally, implementing more stringent controls for environmental factors and ensuring consistency in the visual stimuli presented during the tasks would further strengthen the study’s methodology.

The topic is interesting. The proposed filtering algorithms to enhance eye-tracking data accuracy offer a valuable tool for integrating eye-tracking technology with fMRI applications. The experimental results are encouraging. The manuscript could be considered for publication after minor revisions. Here are some comments:

  1. In Figure 3, it is excellent to see (a3) and (b3) derived from (a2) and (b2). Why? Please provide more explanations.

  2. Figures 4 and 5 are unclear. It is suggested to replace them with two tables respectively.

Reviewer 3 Report

Comments and Suggestions for Authors

This is interesting study on how eye tracking data can be filtered using rather simple techniques. It can be accepted after some revisions as follows:

1. Although the final aim of this research is to use filtered eye tracking data in combination with fMRI the text presents only results on eye tracking data filtering. Motivation of work in relation with MRI data analyses is relevant but the title and abstract should be in line with results reported further in the text.

2. Since there are various filtering techniques a comparison with at least one different approach is a must in order to demonstrate the advantages/disadvantages of the proposed by the authors simple filtering technique.

3. I've noticed and incorrect titles of sub-figures within Figure 3. The legend and obviously the figures are corresponding but on the figure second column should be denoted as "original trajectory" on both rows; the third column has to be entitled "filtered trajectory".

Round 2

Reviewer 1 Report

Comments and Suggestions for Authors

The author has improved the study, making it better. However, another concern should be addressed. The study lacks a comparison with other related studies. Including such comparisons would give a broader context and strengthen the findings.
